# Dysregulation of Placental Lipid Hydrolysis by High-Fat/High-Cholesterol Feeding and Gestational Diabetes Mellitus in Mice

**DOI:** 10.3390/ijms232012286

**Published:** 2022-10-14

**Authors:** Katharina B. Kuentzel, Ivan Bradić, Zala N. Mihalič, Melanie Korbelius, Silvia Rainer, Anita Pirchheim, Julia Kargl, Dagmar Kratky

**Affiliations:** 1Gottfried Schatz Research Center, Molecular Biology and Biochemistry, Medical University of Graz, 8010 Graz, Austria; 2Otto Loewi Research Center, Division of Pharmacology, Medical University of Graz, 8010 Graz, Austria; 3BioTechMed-Graz, 8010 Graz, Austria

**Keywords:** intracellular lipases, mouse placenta, gestational diabetes, maternal high-fat diet, fetal lipid accumulation

## Abstract

Advanced maternal age and obesity are the main risk factors to develop gestational diabetes mellitus (GDM). Obesity is a consequence of the increased storage of triacylglycerol (TG). Cytosolic and lysosomal lipid hydrolases break down TG and cholesteryl esters (CE) to release fatty acids (FA), free cholesterol, and glycerol. We have recently shown that intracellular lipases are present and active in the mouse placenta and that deficiency of lysosomal acid lipase alters placental and fetal lipid homeostasis. To date, intracellular lipid hydrolysis in GDM has been poorly studied despite the important role of FA in this condition. Therefore, we hypothesized that intracellular lipases are dysregulated in pregnancies complicated by maternal high-fat/high-cholesterol (HF/HCD) feeding with and without GDM. Placentae of HF/HCD-fed mice with and without GDM were more efficient, indicating increased nutrient transfer to the fetus. The increased activity of placental CE but not TG hydrolases in placentae of dams fed HF/HCD with or without GDM resulted in upregulated cholesterol export to the fetus and placental TG accumulation. Our results indicate that HF/HCD-induced dysregulation of placental lipid hydrolysis contributes to fetal hepatic lipid accumulation and possibly to fetal overgrowth, at least in mice.

## 1. Introduction

Overweight and obesity are a growing health problem worldwide, and the number of affected individuals is constantly rising. According to the World Health Organization, overweight and obesity are defined as abnormal fat accumulation indicated by a body mass index (BMI) above 25 and 30, respectively. The prevalence varies by geographic location, but overall, 39% of the adult population worldwide is overweight and 13% is obese [1]. More specifically, overweight and obesity are a consequence of increased deposition of triacylglycerols (TG) and are associated with dyslipidemia characterized by changes in circulating lipoproteins. An increased abundance of TG and cholesteryl esters (CE), which affect overall health, can cause problems during pregnancy for both the mother and the developing child. Increased BMI and advanced maternal age were identified as the two main risk factors for gestational diabetes mellitus (GDM) [2,3,4], which is defined as the occurrence of any degree of glucose intolerance during pregnancy [5]. Depending on the ethnicity and geographic location, the prevalence of GDM is between 1% and 30% of all pregnancies (reviewed in [6]). GDM might have adverse effects for the mother, including the development of hypertension, higher risk of infections, cesarean section, and development of type 2 diabetes. The child can also be affected by maternal GDM and develop macrosomia due to hyperglycemia, hyperinsulinemia, and have future risks of childhood diabetes and obesity [6]. This fetal and neonatal programming of obesity due to a pregnancy complicated by GDM results in a vicious circle, as the offspring is more prone to develop obesity and type 2 diabetes later in life. 

During pregnancy, glucose is the primary substrate for the developing fetus [7], and the placenta cannot protect the fetus from glucose oversupply during GDM [8]. Fetal growth is determined by placental nutrient transfer, which is altered in GDM [9]. Other important nutrients besides glucose are fatty acids (FA), which are of vital importance to the developing fetus as major building blocks of membranes, energy substrates, and signaling molecules. To ensure proper fetal development, FA need to be provided by the placenta during pregnancy. Circulating lipids are transported in the form of lipoproteins, which are either (i) hydrolyzed by lipoprotein lipase (LPL) or endothelial lipase (EL) at the placental surface to liberate FA [10] or (ii) taken up by receptor-mediated endocytosis and hydrolyzed intracellularly [11]. Adipose triglyceride lipase (ATGL), hormone-sensitive lipase (HSL), and monoacylglycerol lipase (MGL) catalyze the release of FA from cytosolic TG, DG, MG, and CE [12,13,14], whereas lysosomal acid lipase (LAL) is the sole enzyme known to hydrolyze these lipids in the lysosomal lumen [15], from where FA and free cholesterol (FC) are further exported into the cytosol [16]. We have recently shown that the intracellular CE hydrolases LAL and HSL are expressed and active in the mouse placenta and that LAL deficiency leads to the disturbance of placental lipid metabolism [17]. Transcriptomics and metabolomics analyses revealed altered lipid metabolism in placental cells under high glucose conditions [18] and that placental lipids are the main metabolites with variable abundance in women with GDM [19]. Moreover, placental FA transporters were found to be upregulated in diabetic women [9,20]. A study in 2014 investigated placental gene and protein expression levels of ATGL (encoded by *PNPLA2*), HSL (encoded by *LIPE*), LPL, and EL (encoded by *LIPG*) in women with GDM [21]. Despite increased mRNA expression of *PNPLA2* and its co-activator *ABHD5* and decreased *LIPE* gene expression, protein levels of these lipases were unaltered in women suffering from GDM [21]. Notably, the activity of these enzymes relies on posttranslational modifications and their lipolytic activities in GDM placentae are still unknown. In the human placenta, EL is dysregulated in pregnancies complicated by GDM [22], and placental lipid metabolism is altered due to an abnormal FA profile [18,19,23,24]. However, activity-based studies on intracellular lipases are still missing [25]. 

We therefore investigated whether acid and neutral intracellular lipid hydrolases are dysregulated in a mouse model of maternal high-fat/high-cholesterol (HF/HCD) feeding with and without GDM. Our study revealed that the placentae of dams fed HF/HCD who developed GDM or no GDM accumulated TG due to reduced TG hydrolysis, whereas elevated CE hydrolysis prevented excessive placental CE deposition. Additionally, upregulation of placental cholesterol exporters led to increased cholesterol transport to the fetus, resulting in CE accumulation in the fetal liver and most likely also contributing to fetal overgrowth. 

## 2. Results

### 2.1. Increased Placental Efficiency after HF/HCD Feeding and in GDM

To induce GDM in mice, we combined the two most common risk factors for GDM: obesity and advanced maternal age [2,3,4]. Therefore, mice were challenged for six weeks with a diet enriched with fat and cholesterol (HF/HCD) prior to breeding (Appendix A). To ensure that metabolism is altered upon HF/HCD feeding, we performed a glucose tolerance test (GTT) before mating. Fasting glucose levels of HF/HCD-fed non-pregnant mice remained unaltered (Figure 1A), but glucose clearance was already prolonged compared to chow-fed controls (Figure 1B). Interestingly, HF/HCD feeding did not delay glucose clearance in pregnant mice (Figure 1C). The spontaneous occurrence of GDM was identified by prolonged maternal glucose clearance upon HF/HCD feeding at the end of pregnancy at gestational day (GD) 18, defined as HF/HCD GDM (Figure 1C and Appendix A). Maternal HF/HCD feeding with and without GDM did not affect the number of offspring at GD 19 (Figure 1D). Moreover, placental weight was significantly decreased in all pregnancies on HF/HCD with and without accompanying GDM (Figure 1E), despite increased fetal weight (Figure 1F). Consequently, placentae of HF/HCD and HF/HCD GDM mice were twice as efficient [26] as of chow diet-fed controls (Figure 1G), indicating increased placental nutrient transfer to the fetus. Reduced placental weight prompted us to examine structural markers of the different murine placental zones (junctional zone and labyrinth), which are responsible for the secretion of placental hormones and maternal-fetal exchange, respectively. As expected from the previous results, the expression of genes specific for the junctional zone (*Tpbpa* and *Mash2*) and the labyrinth (*Eomes*) [27] was significantly downregulated (Figure 1H), supporting our hypothesis of a smaller placenta due to HF/HCD feeding. Only the endothelial cell marker *Cd31* was upregulated at the mRNA level in placentae of HF/HCD-fed mice not affected by GDM, whereas the expression of the trophoblast marker *Krt8* remained unchanged (Figure 1I). 

### 2.2. Placentae from Pregnancies Complicated by HF/HCD and GDM Accumulate TG 

To investigate the impact of GDM on placental lipid hydrolase activities, we first determined placental lipid concentrations. Lipid staining with oil red O revealed no significant differences in the labyrinth between chow and HF/HCD without GDM groups but resulted in slightly increased staining in the placenta of mice with GDM (Figure 2A). Biochemical quantification of lipids showed a tendency toward elevated TG levels in HF/HCD without GDM and significantly increased TG concentrations (2.1-fold) in placentae of GDM dams, whereas total cholesterol (TC) and CE concentrations were not affected (Figure 2B). In line with results from our previous study [17], the most abundant lipid in the placenta was cholesterol, more specifically FC (Figure 2B). Increased placental lipid levels prompted us to further examine the occurrence and activities of intracellular lipases. Gene expression levels of neutral (*Pnpla2*, *Lipe*) and acid (*Lipa*) lipid hydrolases in the placenta were upregulated in GDM (Figure 2C). mRNA expression of not only *Pnpla2* but also its inhibitor *G0s2* was increased, whereas the co-activator *Abhd5* and the inhibitor *Hilpda* displayed an unchanged expression profile (Figure 2D). In line with increased placental TG concentrations and upregulated mRNA expression of the endogenous ATGL inhibitor *G0s2*, neutral TG hydrolase activity was significantly reduced in placentae from HF/HCD without GDM mice and tended to be reduced in placentae of GDM dams (Figure 2E). 

### 2.3. Increased Placental CE Hydrolysis in Pregnancies Complicated by HF/HCD and GDM

Despite unchanged placental CE levels, neutral CE and acid TG hydrolase activities were increased and acid CE hydrolase activity tended to be increased upon HF/HCD with and without GDM (Figure 3A,B and Appendix A), in line with elevated mRNA expression levels of *Lipe* and *Lipa*, respectively (Figure 2C). Since LAL is the only enzyme known to hydrolyze TG and CE at an acidic pH [15], we concluded that CE do not accumulate in the placenta upon HF/HCD feeding without and with GDM due to increased hydrolytic activities of neutral and acid CE lipases. Moreover, CE hydrolase activity might be upregulated in the placenta to immediately shuttle cholesterol to the fetus. Gene expression levels of the lysosomal markers lysosome-associated membrane glycoprotein 1 (*Lamp1)* and *Lamp2* remained unaltered, but we observed increased gene transcripts of the lysosomal cholesterol exporter Niemann-Pick disease type C1 (*Npc1)* in placentae of mice after HF/HCD feeding with and without GDM (Figure 3C). Together with increased LAL activity (Figure 3B), these results suggested increased lysosomal cholesterol export into the cytosol due to elevated lysosomal CE hydrolysis. In addition, the placental cholesterol exporters ATP-binding cassette subfamily A member 1 (*Abca1)* and *Abcg1* were upregulated in both conditions (Figure 3D). Moreover, ABCA1 protein expression also increased upon HF/HCD with GDM (Appendix A).

Placentae from mice on HF/HCD without and with GDM did not show increased lipid uptake, but gene expression analysis revealed an upregulation of placental de novo lipogenesis (Figure 3E), which is probably due to prolonged glucose clearance in GDM (Figure 1C). In humans, obesity [28] and GDM [29] are both associated with increased placental and systemic inflammation. Therefore, we also examined these aspects in our mouse model and found that placentae from pregnancies complicated with GDM, but also after HF/HCD feeding without GDM, displayed increased expression levels of inflammatory (*Nlrp3*) and macrophage markers (*Emr1*) (Appendix A). 

### 2.4. Fetal Hepatic Lipid Accumulation after Maternal HF/HCD Feeding and GDM Development

To study the consequences of maternal HF/HCD feeding and the consequent development of GDM on the unborn mouse, we determined hepatic and plasma lipid levels as well as hepatic lipid metabolism-related gene expression in fetuses at GD 19. Although brain weight was higher in HF/HCD-fed and GDM mice (Figure 4A), liver weight remained unaltered (Figure 4B), resulting in a slightly elevated brain-to-liver ratio (a measure to identify intrauterine growth restriction [30]) in offspring from HF/HCD-fed but not GDM mice (Figure 4C). Offspring of GDM dams showed elevated circulating TG but unchanged CE (Figure 4D) and non-esterified FA (NEFA) levels in fetal plasma (Appendix A). Accumulation of TG and CE was more pronounced in fetal livers of HF/HCD and GDM dams (Figure 4E) than in their respective placentae (Figure 2B). In addition, only fetuses of GDM mothers showed increased mRNA expression of hepatic de novo lipogenesis and cholesterol synthesis genes (Figure 4F and Appendix A), most likely due to increased fetal plasma glucose concentrations (Figure 4G). These results suggest that increased fetal hepatic lipid concentrations are not only a consequence of increased lipid synthesis as also seen in HF/HCD fed dams without GDM. Moreover, increased liver lipid accumulation might contribute to fetal overgrowth upon maternal HF/HCD feeding with and without GDM. 

## 3. Discussion

GDM is one of the most common pregnancy disorders worldwide. In 2021, the prevalence of GDM in Europe and North America was 15% and 20.7%, respectively [31]. The mouse model we used to study placental lipid hydrolases in GDM is based on the two major risk factors for the development of GDM: advanced maternal age and overweight/obesity [2,3,4]. To investigate the effects of pre-existing or pregnancy-related maternal obesity, mice were fed a diet enriched with fat and cholesterol prior to and during pregnancy. The prolonged glucose clearance in HF/HCD-fed mice already before mating revealed a metabolic change, which was reversed during pregnancy in the HF/HCD group without GDM but remained prolonged in the GDM group. 

There are conflicting results regarding placental weight in human GDM: Whereas most studies report an increased weight of the placenta (reviewed in [32]), decreased placental weight in diabetic women has also been observed [33]. Thus, the placenta may be both heavier and lighter in GDM. In line with the study from Romero et al. [33], our murine GDM model displayed increased placental efficiency due to smaller placental weight and higher fetal weight. In the present study, we can exclude the possibility of increased fetal size due to smaller litter size [34] in the HF/HCD and GDM groups, as all dams had a similar number of pups per litter, regardless of maternal diet and diabetic condition. In accordance with the reduced placental weight, several structural markers of the placenta, especially for the junctional zone as well as one for the labyrinth, were decreased in the HF/HCD and GDM placentae. A reduced junctional zone in obesity during pregnancy was already shown in a previous mouse study [35]. A similar change in the expression of placental structural markers was observed in another study showing a smaller junctional zone [36]. Elevated expression of the endothelial cell marker *Cd31* further suggested an increased fetal vascularization of the labyrinth in HF/HCD-fed and GDM mice. In summary, our data provide evidence that the placental structure changes with maternal HF/HCD feeding with and without GDM and that increased fetal vascularization in the labyrinth might contribute to enhanced nutrient transport to the fetus. Notably, the observed phenotypes of gestational obesity and GDM are consistent with data from human pregnancies, in which fetal size was also increased in both obesity and GDM [32,37]. 

Despite the maternal diet being enriched with fat and cholesterol, placental CE concentrations remained unchanged. In contrast, placental TG levels were elevated in both HF/HCD without and with GDM groups, mainly due to decreased neutral TG hydrolase activity. This result is in accordance with a study on human placental explants, which reported increased TG content in GDM explants under high glucose conditions [23]. Unexpectedly, we observed increased neutral and acid CE hydrolysis, in line with transcriptional upregulation of the respective lipid hydrolases, suggesting increased CE turnover in HF/HCD and GDM placentae. The enzymes responsible for placental CE hydrolysis are LAL at acidic pH and HSL at neutral pH, the only known CE-degrading lipases expressed and active in the mouse placenta [17]. Increased acid TG lipase activity observed in the present study confirms previous data from streptozotocin-injected rats and diabetic humans [38]. Furthermore, increased placental HSL activity upon HF/HCD without and with GDM, which is in line with a recent publication that showed increased HSL protein expression in the placenta of GDM rats and humans [39], might also be a consequence of diminished ATGL activity to compensate for TG hydrolysis [13]. The observed increased gene expression of *Pnpla2* might be a response to endogenous ATGL inhibition by *G0s2* [40,41]. The other less potent ATGL inhibitor, *Hilpda* [42], plays a neglectable role in placental ATGL inhibition as indicated by its comparable expression in all groups and lower expression in the placenta in general. Moreover, increased expression of cellular cholesterol exporters indicated an enhanced cholesterol transport to the fetus in HF/HCD without and with GDM, consistent with increased fetal vascularization for augmented nutrient transport. Our study re-emphasizes the importance of measuring lipid hydrolase activities because mRNA expression of these enzymes does not necessarily correlate with lipase activity. In this regard, for example, a recent study investigating lipases in human GDM reported unchanged intracellular and extracellular lipolysis by assessing only gene and protein expression of ATGL, HSL, LPL, and EL [21].

Maternal HF/HCD challenge and GDM induced the inflammatory response as well as macrophage marker gene expression in these placentae, in line with data obtained from human placentae of obese women [43]. In addition, maternal HF/HCD feeding and GDM elevated placental de novo lipogenesis gene expression. However, fatty acid synthase (*Fas*) expression was only upregulated in the GDM condition, suggesting increased lipid synthesis due to elevated maternal glucose concentrations, rather than increased placental lipid uptake. Whether FA are more likely to be taken up by placental transporters or to enter the placenta by passive diffusion remains elusive. 

Maternal health status and nutrition also affected fetal lipid metabolism, as fetal circulating TG levels and fetal hepatic TG and CE content were elevated in GDM. Together with unchanged plasma FA levels, this finding indicated that FA of TG or CE are packaged into lipoproteins rather than being directly transported to the fetal circulation. Furthermore, even maternal HF/HCD feeding without GDM resulted in fetal hepatic lipid accumulation. However, lipid synthesis was solely increased in fetal livers of GDM offspring. This suggested that decreased maternal glucose tolerance is directly associated with increased fetal hepatic lipid synthesis due to higher glucose transport to the fetus. These findings clearly indicated that not only glucose is responsible for fetal overgrowth, but also disturbed placental lipid metabolism. This metabolic fuel overload prior to the development of fat depots could be detrimental to the health of the offspring, possibly driving childhood adiposity and early development of non-alcoholic fatty liver disease. 

The mouse model of diet-induced spontaneous onset of GDM is physiologically more suitable to mimic the human situation than diabetes induced by streptozotocin injections, which destroy the pancreatic β-cells and resemble type 1 diabetes [44]. However, a limitation of this model is that the maternal HF/HCD condition which did not result in GDM cannot be directly compared with human obesity, as these mice were only slightly heavier than the control animals. 

Taken together, our data clearly demonstrate that maternal nutrition and health status profoundly influence fetal development. Fetal programming of hyperlipidemia and hepatic lipid deposition, together with higher birth weight, increase the risk for the development of metabolic syndrome, cardiovascular disease, and type 2 diabetes later in life. Intracellular lipolysis is dysregulated in the placenta of HF/HCD-fed dams without and with GDM, resulting in placental TG accumulation, whereas cholesterol is more likely to be immediately exported to the fetus. In addition, disturbances in placental lipid metabolism contribute to altered fetal circulating and hepatic lipid profiles, at least in mice. 

## 4. Materials and Methods

### 4.1. Animals and Sample Collection 

C57BL/6J mice were used as the wild-type mouse strain, as these mice are susceptible to diabetes upon overnutrition [45]. The animals were housed under standard laboratory conditions in a 12 h light/dark cycle and in a clean, temperature-controlled (22 ± 1 °C) environment. Mice had unlimited access to standard chow diet (11.9% caloric intake from fat; 1324, Altromin, Lage, Germany) or HF/HCD (34% crude fat, 1% cholesterol; D12492 (II) mod., +1% cholesterol, Ssniff^®^ Spezialdiäten GmbH, Soest, Germany). To induce GDM, mice were mated after 6 weeks of HF/HCD feeding (reviewed in [46]), and the GDM group was identified on the basis of prolonged glucose clearance [47] measured by GTT at GD 18. Samples were collected at GD 19 and immediately frozen in liquid nitrogen (Appendix A). 

Breeding cages consisted of two females and one male. Four days after breeding, the male mouse was removed, and pregnancy was confirmed by weight gain between days 4 and 12 after breeding (Appendix A). At GD 19, mice were sacrificed by cervical dislocation, and tissues from dams and offspring were collected. After weighing all placentae, the decidua was removed to avoid maternal contamination. All organs were collected and snap frozen. Placental efficiency was calculated as the ratio of birth weight to placental weight [48]. Fetal blood was collected after decapitation and plasma was isolated by centrifugation at 5200× *g* and 4 °C for 7 min. Litters with less than five offspring were excluded from all experiments and analyses [49] to avoid increased fetal weight due to smaller litter size [34]. All experiments were performed with at least three samples from different pregnancies in accordance with the European Directive 2010/63/EU and approved by the Austrian Federal Ministry of Education, Science and Research (Vienna, Austria; BMWFW-66.010/0165-V/3b/2019).

### 4.2. Glucose Tolerance Test

The mice were fasted for 6 h prior to basal glucose measurement with a glucometer (AccuCheck, Roche Holding AG, Basel, Switzerland). Afterwards, 2 g glucose per kg body weight were injected intra-peritoneally. Blood glucose from the tail vein was determined in the following intervals post-injection: 15, 30, 60, 90, 120, 180 min. Data are presented as fold change over basal glucose concentration [50].

### 4.3. Plasma Parameters

Concentrations of TG, TC, and FC were determined in duplicate of 2 µL plasma using enzymatic kits following the manufacturer’s protocol (DiaSys, Holzheim, Germany). CE concentrations were calculated by subtracting FC from TC.

Plasma concentrations of NEFA were determined in duplicate of 5 µL plasma with an enzymatic kit following the manufacturer’s description (NEFA-HR; ©FUJIFILM Wako Pure Chemical Corporation, Takasaki, Japan). 

Plasma glucose levels were determined in duplicate of 5 µL fetal plasma with a glucometer (AccuCheck, Roche Holding AG, Basel, Switzerland).

### 4.4. RNA Isolation, cDNA Preparation and Real-Time PCR

Total RNA was isolated with TRIsure^TM^ reagent according to the manufacturer’s protocol (Meridian Bioscience, Cincinnati, OH, USA). Two micrograms of RNA were reverse transcribed using the High capacity cDNA reverse transcription kit (Applied Biosciences, Carlsbad, CA, USA), diluted 1:50 and 3 µL of cDNA was used for quantitative real-time PCR. All samples were analyzed in duplicate and normalized to the housekeeper gene *cyclophilin A*. Primers used for analysis are listed in Table 1. 

### 4.5. Oil Red O Staining

Paraformaldehyde-fixed tissues were transferred to a 30% sucrose solution prior to cryosectioning. Sections (5 µm) were cut and rehydrated in PBS. For staining with ORO (Sigma-Aldrich, St. Louis, MO, USA), slides were incubated for 1 h in the freshly prepared ORO staining solution (1.4 g ORO in 200 mL 1,2 Propanediol) and afterwards mounted using Dako mounting medium. Slides were visualized with an Olympus BX63 microscope and images were taken using an Olympus DP73 camera (Olympus, Shinjuku, Japan). 

### 4.6. Lipid Extraction

Tissues in lysis buffer (100 mM potassium phosphate, 250 mM sucrose, 1 mM EDTA, pH 7) were sonicated twice for 10 s on ice, and protein amount was determined after centrifugation. Lipids were extracted from 1 mg of protein by the Folch’s method [51]. TG, TC, and FC concentrations were determined as described above. 

### 4.7. CE and TG Hydrolase Activity Assays

Enzymatic activities were assessed as described previously [17,52]. Briefly, to determine acid CE and TG hydrolase activities, tissues were lysed in acid citrate buffer (containing 54% 100 mM citric acid monohydrate and 46% 100 mM trisodium citrate, dehydrated, pH 4.2). Neutral CE and TG hydrolase activities were measured in samples lysed in lysis buffer pH 7 containing 1 mM DTT and protease inhibitor cocktail 1:1000.

### 4.8. Western Blotting

Placenta samples were sonicated twice for 10 s on ice in lysis buffer and protein concentrations were determined in the supernatant after centrifugation for 10 min at 1000× *g* and 4 °C using the DC™ Protein Assay Kit (Bio-Rad Laboratories, Hercules, CA, USA). Fifty micrograms of protein were separated by SDS-PAGE and transferred to a PVDF membrane. The following anti-mouse antibody was used: ABCA1 (ab18180, 1:1000, Abcam, Cambridge, UK), and PonceauS as a loading control. 

### 4.9. Statistical Analysis

Statistical analysis was performed using GraphPad Prism software 5.6. Significance was calculated by unpaired Student’s *t*-test or 1-way analysis of variance (ANOVA) followed by Bonferroni post hoc test. Data are shown as mean + or ± SD. For real-time PCR analysis, the 2^−ΔΔCT^ method was used. The following significance levels were used: * *p* < 0.05, ** *p* ≤ 0.01, *** *p* ≤ 0.001 for comparison to the chow condition and ^#^ *p* < 0.05 for the comparison of HF/HCD w/o GDM and HF/HCD GDM.

## Figures and Tables

**Figure 1 ijms-23-12286-f001:**
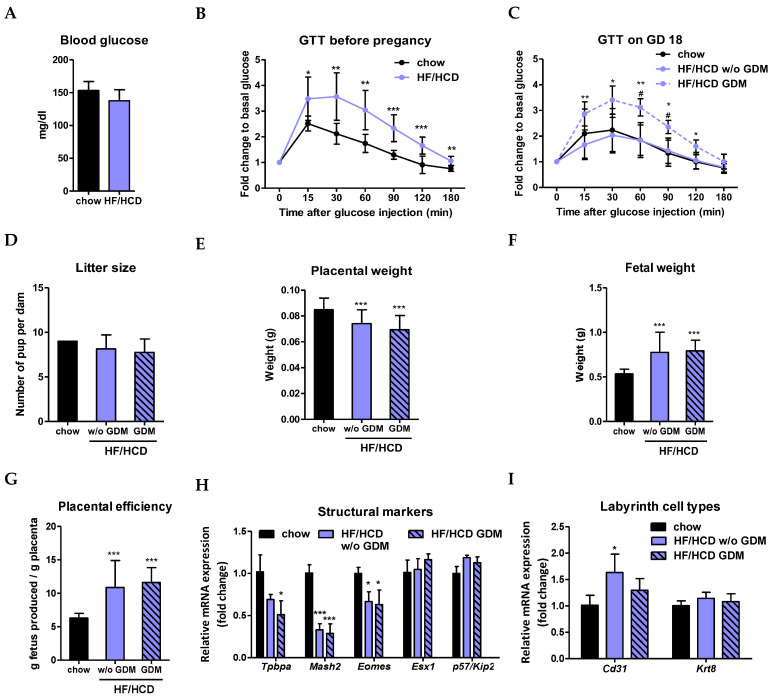
Increased placental efficiency after HF/HCD feeding and in GDM at the end of pregnancy. (**A**) Blood glucose levels after 6 h of fasting and (**B**) glucose tolerance test (GTT) before pregnancy (*n* = 3 chow, 9 HF/HCD). (**C**) GTT at gestational day (GD) 18 (*n* = 3 chow, 8 HF/HCD w/o GDM, 4 HF/HCD GDM). (**D**) Number of pups per dam indicated as litter size (*n* = 3 chow, 8 HF/HCD w/o GDM, 4 HF/HCD GDM). (**E**) Placental weight, (**F**) fetal weight and (**G**) placental efficiency at GD 19 (*n* = 27 chow, 69 HF/HCD w/o GDM, 24 HF/HCD GDM). (**H**) mRNA expression of placental markers for the junctional zone and the labyrinth (*n* = 4). (**I**) mRNA expression of *Cd31* (endothelial cell marker) and *Krt8* (trophoblast marker) in the labyrinth (*n* = 4). Data represent mean ± SD and were analyzed with 1-way ANOVA followed by Bonferroni post hoc test. * *p* < 0.05, ** *p* ≤ 0.01, *** *p* ≤ 0.001, indicating the comparison to the chow condition and ^#^ *p* < 0.05 the comparison between HF/HCD w/o GDM and HF/HCD GDM.

**Figure 2 ijms-23-12286-f002:**
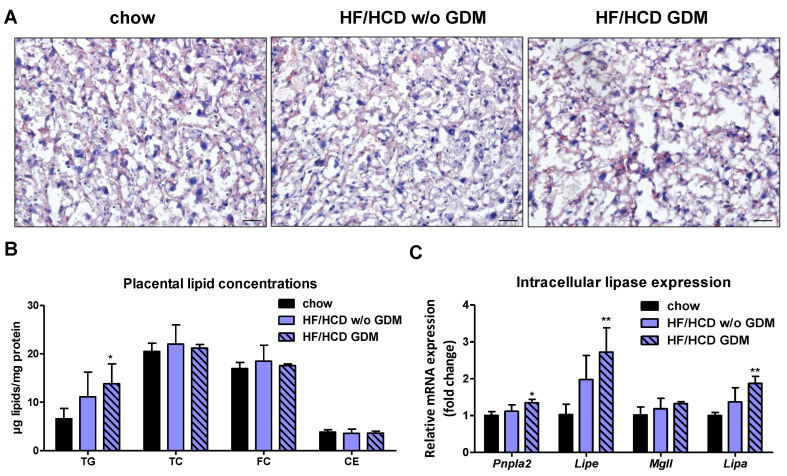
Decreased neutral TG hydrolase activity in placentae of HF/HCD-fed and GDM mice. Placental (**A**) oil red O staining and (**B**) lipid concentrations of chow, HF/HCD w/o GDM, and HF/HCD GDM mice (*n* = 4–5). Scale bar, 100 µm. Gene expression of (**C**) lipases and (**D**) endogenous ATGL co-factors in placentae of these mice (*n* = 4). (**E**) Placental TG hydrolase activity at neutral pH (*n* = 5). Data represent mean + SD and were analyzed with 1-way ANOVA followed by Bonferroni post hoc test. * *p* < 0.05, ** *p* ≤ 0.01, indicating the comparison to the chow condition.

**Figure 3 ijms-23-12286-f003:**
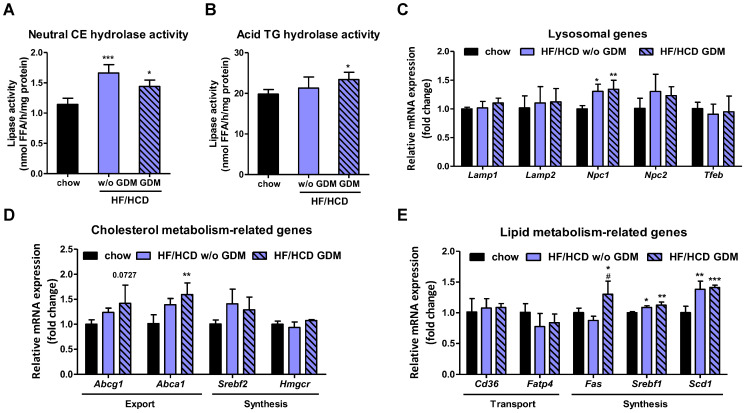
Increased placental cholesterol export in GDM. (**A**) Neutral CE hydrolase activity and (**B**) acid TG hydrolase activity in placentae of chow, HF/HCD w/o GDM, and HF/HCD GDM dams (*n* = 4–5). Gene expression levels of (**C**) lysosomal, (**D**) cholesterol export- and synthesis-related genes, and (**E**) lipid-metabolism-related genes in placentae of these mice (*n* = 4). Data represent mean + SD and were analyzed with 1-way ANOVA followed by Bonferroni post hoc test. * *p* < 0.05, ** *p* ≤ 0.01, *** *p* ≤ 0.001, indicating the comparison to the chow condition and ^#^ *p* < 0.05 the comparison between HF/HCD w/o GDM and HF/HCD GDM.

**Figure 4 ijms-23-12286-f004:**
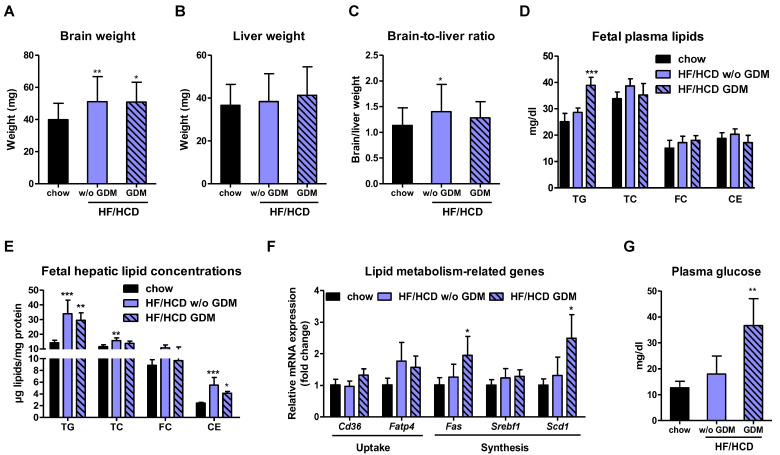
Elevated plasma and hepatic lipid concentrations in offspring of HF/HCD-fed dams without and with GDM. (**A**) Fetal brain weight, (**B**) liver weight, and (**C**) brain-to-liver ratio. Fetal (**D**) pooled plasma lipid concentrations (*n* = 3 chow, 6 HF/HCD w/o GDM, 3 HF/HCD GDM), (**E**) hepatic lipid content (*n* = 5), and (**F**) hepatic gene expression of lipid metabolism-related genes (*n* = 4). (**G**) Plasma glucose concentrations in pooled fetal plasma samples (*n* = 3 chow, 6 HF/HCD w/o GDM, 3 HF/HCD GDM). Data represent mean + SD and were analyzed with 1-way ANOVA followed by Bonferroni post hoc test. * *p* < 0.05, ** *p* ≤ 0.01, *** *p* ≤ 0.001 indicates the comparison to the chow condition.

**Table 1 ijms-23-12286-t001:** Primers used for real-time PCR.

Gene	Forward Sequence 5′-3′	Reverse Sequence 5′-3′
*Abca1*	CTCTTCATGACTCTAGCCTGGA	ACACAGACAGGAAGACGAACAC
*Abcg1*	CTTTCCTACTCTGTACCCGAGG	CGGGGCATTCCATTGATAAGG
*Abhd5*	GGTTAAGTCTAGTGCAGC	AAGCTGTCTCACCACTTG
*Cd31*	ACCGGGTGCTGTTCTATAAGG	TCACCTCGTACTAATCGTGG
*Cd36*	GCAGGTCTATCTACGCTGTG	GGTTGTCTGGATTCTGGAGG
*Cyclophilin A*	CCATCCAGCCATTCAGTCTT	TTCCAGGATTCATGTGCCAG
*Emr1*	CTTTGGCTATGGGCTTCCAGTC	GCAAGGAGGACAGAGTTTATCGTG
*Eomes*	GCGCATGTTTCCTTTCTTGAG	GAAGCGCCAGTGGTTAGGG
*Esx1*	CCCATGCATCCTCAAATGATG	GCCTAAATGGTGGAGGCATTC
*Fas*	GAAGCCGAACACCTCTGTGCAGT	GCTCCTTGCTGCCATCTGTATTG
*Fatp4*	GGCACAGACACTCACTGGAC	TGCGGTTTTCCATAAAGAGGG
*G0s2*	GCCACCGAATCCAGAACTGA	TTGATTGCTCGCACAGCCTA
*Hilpda*	TCCGTGACTCCCCGAGAA	GCCCAGCACATAGAGGTTCA
*Hmgcr*	TGTTCACCGGCAACAACAAGA	CCGCGTTATCGTCAGGATGA
*Krt8*	CAAGGTGGAACTAGAGTCCCG	CTCGTACTGGGCACGAACTTC
*Lamp1*	CAGCACTCTTTGAGGTGAAAAAC	CCATTCGCAGTCTCGTAGGTG
*Lamp2*	TGTATTTGGCTAATGGCTCAGC	TATGGGCACAAGGAAGTTGTC
*Lipa*	GCTGGCTTTGATGTGTGGATG	ATGGTGCAGCCTTGAGAATGA
*Lipe*	GCTGGTGACACTCGCAGAAG	TGGCTGGTGTCTCTGTGTCC
*Mash2*	AACCGCGTAAAGCTGGTAAACT	TCTCCACCTTACTCAGCTTCTTGTT
*Mgll*	CGGACTTCCAAGTTTTTGTCAGA	GCAGCCACTAGGATGGAGATG
*Nlrp3*	TCGCAGCAAAGATCCACACAG	ATTACCCGCCCGAGAAAGG
*Npc1*	AATGCCTGCCGTGATGTG	CGCTTGTCCGTTGTCTTTATTG
*Npc2*	GTCAACATCACCTTTACC	GATTCCACTCTTACAACC
*Pnpla2*	GCCACTCACATCTACGGAGC	GACAGCCACGGATGGTGTTC
*Scd1*	CCGGAGACCCCTTAGATCGA	TAGCCTGTAAAAGATTTCTGCAAACC
*Srebf1*	CAAGGCCATCGACTACATCCG	CACCACTTCGGGTTTCATGC
*Srebf2*	TGAAGGACTTAGTCATGGGCAC	CGCAGCTTGTGATTGACCT
*Tfeb*	AAGGTTCGGGAGTATCTGTCTG	GGGTTGGAGCTGATATGTAGCA
*Tpbpa*	GGAGTGGCCTCAGCTGCTAT	AACTTCTTTATCCTTCTGCTCTTGCA

## Data Availability

The data presented in this study are available upon request from the corresponding author.

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
