# Peer review of "Dysregulation of Placental Lipid Hydrolysis by High-Fat/High-Cholesterol Feeding and Gestational Diabetes Mellitus in Mice"

_ijms, 2022, doi:10.3390/ijms232012286_

Round 1

Reviewer 1 Report

Study by Katharina B. Kuentzel et al ‘Dysregulation of Placental Lipid Hydrolysis by High-fat/high-cholesterol Feeding and Gestational Diabetes Mellitus in Mice’.

The authors provide evidence for high-fat/high-cholesterol (HF/HCD) induced dysregulation of placental lipid hydrolysis contributes to fetal hepatic lipid accumulation and possibly to fetal overgrowth in mice.

Authors provided evidence that intracellular lipases are dysregulated in pregnancies complicated by maternal high-fat/high-cholesterol HF/HCD feeding with and without gestational diabetes mellitus (GDM). Authors explained that the placenta of HF/HCD-fed mice with and without GDM were more efficient, indicating increased nutrient transfer to the fetus. Cholesteryl esters (CE) increased activity in placenta fed HF/HCD with or without GDM was responsible for increased cholesterol export to the fetus and placental triacylglycerol (TG) accretion.

 It is well known that obesity and overweight is a growing problem worldwide causing myriads of health problems (39% of the adult population worldwide is overweight and 13% is obese). Deposition of triacylglycerols and associated dyslipidemia characterized by changes in circulating lipoproteins is the cause of obesity. Authors mentioned that enhanced TG and CE can cause problems during pregnancy for both mother and the developing child and Increased BMI and advanced maternal age are the 2 main risk factors for gestational diabetes mellites (DM) is responsible for maternal hypertension, infections, cesarean section and type 2 DM and child develop macrosomia and future enhanced risk of diabetes and obesity. Authors further explained that placental nutrient transfer determines the fetal growth which is altered in GDM. Glucose are fatty acids (FA), which are vital for developing fetus. Authors suggested that circulating lipids are transported in the form of lipoproteins, which are either hydrolyzed by lipoprotein lipase (LPL) or endothelial lipase (EL) at the placental surface to liberate FA or taken up by receptor-mediated endocytosis and hydrolyzed intracellularly. Authors stated that transcriptomics and metabolomics analyses revealed altered lipid metabolism in placental cells under high glucose conditions and placental lipids are the main metabolites with variable abundance in females with GDM. Authors with their study observed that   placentae fed HF/HCD with and without GDM accumulated TG due to reduced TG hydrolysis and increased CE hydrolysis prevented excessive placental CE deposition causing increased cholesterol transport to the fetus, resulting in fetal overgrowth.

 Authors induced GDM in mice with diet rich with fat and cholesterol (HF/HCD) prior to breeding and found prolonged glucose clearance compared to chow-fed controls. Authors found that placentae of HF/HCD and HF/HCD GDM mice were twice as efficient as of chow diet-fed controls indicating increased placental nutrient transfer to the fetus. Authors further found that Biochemical quantification of lipids showed a tendency toward elevated TG levels in HF/HCD without GDM and significantly increased TG concentrations in placenta of GDM and reduced TG hydrolase activity significantly. Authors further found that CE does not accumulate in the placenta upon HF/HCD feeding without and with GDM due to increased hydrolytic activities of neutral and acidic CE lipases. Increased liver lipid accumulation might contribute to fetal overgrowth upon maternal HF/HCD feeding with and without GDM.

Finally, authors concluded that maternal nutrition and health (BMI) greatly influence fetal development. Disturbances in placental lipid metabolism and Intracellular lipolysis are dysregulated in the placenta of HF/HCD-fed without and with GDM, resulting in placental TG accumulation, resulting in cholesterol immediate transport to the fetus.

In conclusion, the manuscript is well presented and well written. The tables and graphs in the manuscripts are clear and have a direct correlation with the aim of the study. The study can be reproducible, references used are most recent and no abnormal number of self-citations were noted. The manuscript seems scientifically sound and experimental design is appropriate to test the hypothesis. The discussion and conclusion have enough evidence and arguments in relation to the aim of the study.
The paper can be accepted to publish.

Reviewer 2 Report

Thank you for the manuscript with interesting results. 

1. Why is the abbreviation WHO introduced when you do not use it except from once in the Introduction?

2. Why did you choose 6 hours of fasting i rats when women undergo 8-12 hours of fasting prior to OGTT?

3. Which criteria are used to diagnose GDM? 

4. Could you choose a better reference to show that age is important with regard to the development of GDM?

5. Could you please show exact glucose levels on the OGTT? It is hard to see how sick the rats have become?

6. Did any of the rats have 2hPG > 11.1 mmol/L suggesting new-onset DM? And if so were the results different in these rats?

7. Please use af reference for the first statement in the Discussion. 

Reviewer 3 Report

In this article, the authors used high-fat/high-cholesterol (HF/HCD) diet to treat mice before pregnancy, and then tested “prolonged maternal glucose clearance” to identify HF/HCD induced GDM mice. They found decreased placental weight and increased fetal weight in HF/HCD-fed GDM group. In HF/HCD-fed GDM group, mice showed both increased TG concentration with decreased neutral TG hydrolase activity and increased placental cholesterol export due to increased placental CE hydrolysis. In the offspring of HF/HCD-fed GDM, fetal showed elevated plasma and hepatic lipid concentrations. 

Comments:

1.     Does “placental efficiency (g fetus produced / g placenta)” have any other index because the current calculation seems weak to claim increased efficiency of placental nutrient transfer to the fetus?

2.     In line 98, the author used “prolonged maternal glucose clearance” as an index to identify HF/HCD GDM, how to set the threshold value? 

3.     In Figure 1H, it might be weak to use only two structural marker genes to prove structural alteration in the junctional zone and labyrinth of placenta.

4.     What is the meaning underlying “Brain-to-liver ratio” showed in Figure 4C?

5.     The gene names of human in line 70 should be capitalized.

Round 2

Reviewer 2 Report

Thank you for the revised manuscript, I have no further comments. 

Author Response

Thank you very much for your positive reply.